# Intranasal Coronavirus SARS-CoV-2 Immunization with Lipid Adjuvants Provides Systemic and Mucosal Immune Response against SARS-CoV-2 S1 Spike and Nucleocapsid Protein

**DOI:** 10.3390/vaccines10040504

**Published:** 2022-03-24

**Authors:** Anirban Sengupta, Mohammad Azharuddin, Maria E. Cardona, Claudia Devito, Eleanore von Castelmur, Anna Wehlin, Zuzanna Pietras, Maria Sunnerhagen, Robert Selegård, Daniel Aili, Ali Alamer, Jorma Hinkula, Noha Al-Otaibi

**Affiliations:** 1Department of Biomedical and Clinical Sciences, Linköping University, 58185 Linköping, Sweden; anirban.sengupta@liu.se (A.S.); mazharuddin@cdtltd.co.uk (M.A.); suecia_claudia@hotmail.com (C.D.); 2Laboratory of Molecular Materials, Division of Biophysics and Bioengineering, Department of Physics, Chemistry and Biology, IFM, Linköping University, 58183 Linköping, Sweden; marca07@liu.se (M.E.C.); eleanore.voncastelmur@liu.se (E.v.C.); anna.wehlin@liu.se (A.W.); zuzanna.pietras@liu.se (Z.P.); maria.sunnerhagen@liu.se (M.S.); robert.selegard@liu.se (R.S.); daniel.aili@liu.se (D.A.); 3King Abdulaziz City for Science and Technology (KACST), Riyad 11442, Saudi Arabia; aaalamer@kacst.edu.sa (A.A.); naalotaibi@kacst.edu.se (N.A.-O.)

**Keywords:** pandemic, COVID-19, immunity, vaccine, formula, antigen, adjuvants

## Abstract

In this preclinical two-dose mucosal immunization study, using a combination of S1 spike and nucleocapsid proteins with cationic (N3)/or anionic (L3) lipids were investigated using an intranasal delivery route. The study showed that nasal administration of low amounts of antigens/adjuvants induced a primary and secondary immune response in systemic IgG, mIL-5, and IFN-gamma secreting T lymphocytes, as well as humoral IgA in nasal and intestinal mucosal compartments. It is believed that recipients will benefit from receiving a combination of viral antigens in promoting a border immune response against present and evolving contagious viruses. Lipid adjuvants demonstrated an enhanced response in the vaccine effect. This was seen in the significant immunogenicity effect when using the cationic lipid N3. Unlike L3, which showed a recognizable effect when administrated at a slightly higher concentration. Moreover, the findings of the study proved the efficiency of an intranasally mucosal immunization strategy, which can be less painful and more effective in enhancing the respiratory tract immunity against respiratory infectious diseases.

## 1. Introduction

The fast spread of the new strain of SARS-CoV-2 (COVID-19) compromised global public health. COVID-19 typically causes mild respiratory infections with typical symptoms, such as fever, dry cough, and headache. In vulnerable individuals, more severe symptoms and increased mortality are seen [1,2]. Previous viral infectious epidemics and pandemics have been occasionally prevented or curbed through massive vaccination campaigns, such as in the case of Ebola and influenza A. In recent years, vaccine discoveries have evolved, revealing a number of high-potential novel vaccine candidates, which have been studied and taken through preclinical and clinical trials to become novel vaccines to prevent or mitigate the associated symptoms of the disease. Consequently, several vaccine platforms are available, including SARS-CoV-2 vaccines, such as plasmid DNA vaccines, messenger RNAs, recombinant viral vector vaccines and inactivated recombinant protein, epitopes, and adenovirus-based vectors [3,4,5,6].

The recent direction of vaccine design and formulation research is heading toward more attractive targets, that can achieve high efficiency, safety, and, potentially, could target a few viruses at once. For instance, the S protein has been shown to have a key role in inhibiting viral infection, as well as humoral protection and cell-mediated immune response in the hosting body during infection since. It mediates receptor recognition, cell attachment, and fusion during viral infection, not only in COVID-19 [7,8,9] but also in other viruses, such as Ebola, influenza, and paramyxovirus [10]. In trial studies, the recombinant S1-protein evidently induced a strong immunogenic response [11].

The nucleocapsid (NC) is another important protein for the viral RNA-binding and genome packaging [12,13]. Using an NC protein in vaccines could potentially improve their protection against viral infectious diseases. It has been used in pre-clinical studies, such as in Crimean–Congo hemorrhagic fever virus [14]. Most recently, an NC protein application in testing the immunoreactivity of patients showed a high and specific immune response [15]. Therefore, it is a tempting agent to apply in vaccinations.

The use of vaccines adjuvants is paramount in successful vaccine design and strategy to enhance sufficient humoral and cellular immune responses against antigens. Adjuvants, such as aluminum hydroxide, aluminum phosphate, and liposomes, are commonly used in vaccine formulation. In particular, liposomes or phospholipid-based vesicles, which have been used for decades, are appealing adjuvants and vehicles as they are abundant in mucosal tissues and play a vital role in cellular function and architecture. Additionally, antigens trapped in liposomes proved to enhance immunogenicity responses [16]. Additionally, the delivery mode of the vaccines is considerably important for successful vaccination. The most common approaches are the subcutaneous and/or intramuscular injections, which are associated with pain, stress for recipients, plus the requirement of sufficient training for personnel to be able to administrate them. Moreover, they carry a risk of transmitting diseases, which potentially occur during reuse of contaminated needles or via needle-stick injuries. There are alternative routes, which are sufficient for vaccine delivery, such as oral vaccination (e.g., rotavirus, typhoid, and polio vaccines) and intranasal vaccination (e.g., live attenuated influenza vaccine). Until now, nasal spray vaccination use is only limited to live attenuated viruses [17].

In this study, we explore the effect of the recombinant proteins SARS-CoV-2, S1 spike and the NC protein to target different viral proteins. The aim of our study is to determine the potential vaccine use and adjuvant-mediated enhancement of its efficacy in the wild-type mouse model. Our aim with using conventional mice first was to identify how immunogenic our vaccine candidates are along with the adjuvants added to the antigens. Furthermore, to immunize healthy and common research lab mice, provide reagents, such as serum IgG and IgA so that future research may allow for detailed immunological studies and passive immunotherapy study materials for animals that demonstrate an in vivo viral challenge.

As a growing number of mouse-adapted SARS-CoV-2 viruses are tried on for infecting the wild-type laboratory mice, it is important to study the potential immune responses generated within them. Before performing a challenge, it is valuable to study if the selected vaccine candidates and adjuvants are sufficiently immunogenic in mice to provide meaningful challenge studies. The study also provides initial safety data and may identify potential unexpected side effects with the vaccine or adjuvant components. The study aims to examine the immunogenicity effects of antigens with anionic and cationic lipid adjuvants, when administrated intranasally, on the systemic and mucosal humoral immune response to the antigens. 

## 2. Materials and Methods

Female mice, 6–8 weeks old C57Bl were obtained from Janevier, Sollentuna, Sweden.

Human endogenous adjuvant L3 (anionic) and a derivative of the N3 (cationic) lipid formulations were obtained from Merck (Darmstadt, Germany). Spike protein and the recombinant NC protein were purchased from SinoBiological (SCG, Toronto ON, Canada/Sinobiologicals, Eschborn, Germany). For cells culture, DMEM plus 2 mM L-glutamax and Na-pyruvate was used. Penicillin-streptomycine and a 10% inactivated bovine serum albumin (BSA) were added to the medium. The positive control used was obtained from SARS-CoV-2 infected individuals. The negative control serum was pre-infection serum. The medium, penicillin-streptomycin, BSA and PFA were purchased from Sigma-Aldrich/Merck (Solna, Sweden). 

### 2.1. Experimental Design

The mice were assigned to eight experimental groups (*n* = 5 each group), including two control groups. One of the control groups was infected with the noninfectious virus (SARS-CoV-2-Iso_LiU-Human-2020-03-04-Swe; positive control) and the other one was given the same amount of saline solution (negative control). A different formulated vaccine was administrated by intranasal spray to each study group (Table 1). The animals received a standard rodent diet. Food and water were available *ad libitum*. Water-supply was changed daily. The cages were changed weekly. All experiments were performed according to the ethical permissions and guidelines of the Swedish national ethical board and the national animal health care rights.

#### 2.1.1. Vaccines and Adjuvants Formulation and Administration Procedure 

The L3 and the N3 lipid adjuvant formulations were used at 1.5% (*w*/*v*) by mixing each with either 0.1 or 1 µg of S1 recombinant Spike protein and 0.1 or 1 µg of the recombinant NC protein in 10 µL volume/mouse.

The anionic L3 adjuvant was prepared as previously described [18,19,20]. Briefly, 8% L3 was prepared by blending oleic acid (0.465 g) with oleic lauric acid (0.345 g), gently heated, thereafter sonicated with 9.2 mL of 0.1 M Tris-buffer (pH 8.0) and the pH was adjusted with 5 M NaOH to 8.0. The 8% lipid emulsions were then mixed with the S1 spike and NC antigens in order to obtain a 1.5% or 2% final lipid concentration in the prepared vaccines.

The cationic N3 adjuvant was prepared as previously described [21] by mixing 0.4 g of 1:1 (molar ratio) oleylamine and mono-olein, mixed with 9.6 mL of 0.1 M Tris-buffer, pH 8.0, in a test tube. After that, the emulsion of N3 was sonicated for two minutes to obtain the emulsion and, finally, the pH was adjusted to 8.0. The vaccine formula with the S1 spike protein and the NC were mixed with the N3 emulsion at 1:1 (*v*/*v*) to make a final concentration of 1.5% N3 lipid formulation.

The formulated vaccines were administrated to the sedated mice by nasal spray in a total of two doses over 21 days. The sedation procedure has been performed as previously described [22,23,24]. The immune stimulant solution was given as a single dose on day 0 and day 21 [23,24]. The vaccine administration was performed in 3 steps. Mice were anesthetized with isoflurane in a gas chamber and queued for administration. Each time a single mouse was taken out of the chamber, held in a supine position, then given a nasal spray of 5 µL nostril antigen/adjuvant using a 1–10 µL pipette and laid back inside the gas chamber in a supine position. Animals were checked daily in their cages.

#### 2.1.2. Serology Screening 

Blood samples (1.5 mL) were collected from the experiment animals on days 14 and 42. The serum (0.15 mL) was separated following the standard procedure described by Hoff J. 2000. For the serum screening, 96-well microplates (Nunclon, Copenhagen, Denmark) were coated with 0.5 µg/mL recombinant S1 and NC proteins in sterile PBS and incubated overnight at 4 °C. The serum samples were diluted in PBS 2.5% fat-free milk buffer with 0.05% Tween 20. Serial dilutions from 1:100 to 1:100,000 were prepared and 100 µL from each dilution were transferred in duplicate wells into antigen-coated plates, followed by incubation at 37 °C for 90 min. Thereafter, the plates were rinsed with PBS+0.05% Tween 20 (PBS-T). The HRP-labelled conjugate goat-anti-mouse IgG and IgA (BioRad, Richmond, CA) was added to each well (100 µL/well) and incubated at 37 °C for 90 min. Then, plates were washed again with PBS-T, followed by the addition of 0.2 g o-phenylenediamine (OPD) to 0.03% H_2_O_2_ in 1 ml. Then, plates were kept in the dark at room temperature (RT) for 30 min. The reaction was terminated by adding 100 µL of 2.5% H_2_SO_4_ to each well and the absorbance was measured at OD_490_. The cut-off value for positive reactivity was calculated from the mean OD_490_ plus 3SD for negative control samples.

### 2.2. Microneutralization Assay (MNA)

Serum MNA was performed as previously described [25,26,27]. In brief, 75 µL of serum were dispensed into a flat well cell culture plate in duplicate. Virus (SARS-CoV-2-Iso_LiU-Human-2020-03-04-Swe) was added to each well at a concentration of 100–130 PFU/mL and incubated for 1 h at 37 °C and 5% CO_2._ After incubation, the serum/virus mixture was placed into wells with 5 × 10^4^ Vero E6 cells. The plates were kept at 37 °C and 5% CO_2_ for 96 h before being examined under the microscope for the ratio of healthy cells vs. cytopathic effect (CPE). Cells were washed post-incubation and fixed with 4% PFA for 30 min, followed by staining with 0.1% crystal violet for 4–5 min. Diluted serum showing 50% inhibition of virus-induced CPE was considered as the MNA titer. Neutralizing antibodies titers were expressed as serum dilutions resulting in a 50% inhibiting neutralizing titer in vitro.

### 2.3. Cell-Mediated Immunity

For the murine IL-5 and IFN-gamma determinations, the ELISpot capture assay was used. ELISpot assays were performed with murine IL-5 and IFN-gamma capture assays with spleen cells after the second immunization, following the recommendations from the manufacturer (MabTech, Nacka, Sweden). In brief, single-cell suspensions were prepared, washed and added to microplate wells (200,000 cells/well) in duplicate. The DMEM with 2 mM L-glutamax, 1% penicillin-streptomycine, 4 mM Na-pyruvate, and 10% inactivated bovine serum (Sigma-Aldrich) without antigens was used as a negative control and Concanavalin A (5 µg/mL) as a positive control. Recombinant SARS-CoV-2 antigens were added as stimulating antigens at a pool of SARS-antigen concentration (1 µg/mL), with S1 spike protein, S1 receptor-binding motif (S1 RBM) recombinant E.coli and synthetic peptide antigen and nucleoproteins (NC) as antigens.

### 2.4. Statistical Analyses

Data analysis was performed using GraphPad Prism (San Diego, CA, USA). Comparisons between study groups were performed with non-parametric methods using the Mann–Whitney U test. *p* < 0.05 was considered statistically significant.

## 3. Results

### 3.1. Serology

The results showed that all SARS-CoV-2 recombinant protein immunized animals, which received two doses, developed antibodies against the administrated SARS-antigens in comparison to the negative control. The magnitude of the serological response varied significantly between the study groups as shown in Figure 1A,B. Mice fully immunized with 2 doses of recombinant antigens mixed with cationic adjuvants (N3) showed significantly higher serum IgG titers in comparison to the anionic adjuvants (L3). The difference in increase was almost 10-fold. Notably, a slightly higher concentration of L3 adjuvants (2%) in the vaccine formula showed a comparable level of antibodies titration to what was seen in the presence of N3.

A similar pattern was seen in IgA titration post the second dose in which S1 + NC with cationic adjuvants N3 or 2% of anionic adjuvants L3 showed a significant increase. These were only observed post introducing the second dose of the formulated vaccines which was almost 10-fold higher than the first dose in each group. 

### 3.2. Microneutralization for SARS-CoV-2 Antibodies

Microneutralizing antibody titers were variable between samples obtained from different treated groups. Antibodies titration obtained from groups 5 and 6, which received the S1 spike + NC protein shielded by N3 cationic lipid, were highly significant. Whereas group 4, which received the same antigens but with anionic lipid L3, was slightly higher than the positive control group (Figure 2).

### 3.3. Mucosal Immunity

For humoral immune responses at local mucosal sites, all SARS-CoV-2 immunized animals, with recombinant proteins and adjuvant, developed nasal/respiratory and gastrointestinal tract IgA antibodies against the SARS-antigens. The differences in IgA titers were significant between study groups (Figure 3). Mice immunized with recombinant antigens mixed with cationic adjuvants responded with significantly higher mucosal IgA titers. The median and range are shown in the box graphs.

### 3.4. Spleen Cell Murine IL-5 and IFN-Gamma Responses In Vitro

All SARS-CoV-2 immunized animals with recombinant proteins with adjuvant, developed detectable IL-5 secreting spleen cells upon in vitro stimulation with recombinant antigens. Significantly higher levels of IL-5 secretion were seen in animals receiving antigens in cationic adjuvant N3 as compared with the negative control (Figure 4A). The median and range are shown in the box graphs.

The IFN-gamma secreting spleen cells were calculated in spleen cells from the same mice and stimulated with the same antigens (Figure 4B). Significantly higher levels of IFN-gamma secretion were seen in animals receiving antigens in a cationic adjuvant.

## 4. Discussion

The obtained data in the presented study showed that the SARS-CoV-2 nasal vaccination formula, composed of mucosal adjuvants and recombinant coronavirus SARS-CoV-2 proteins S1 and NC at a low concentration, induces a mucosal and systemic humoral immune response. The immunogenicity effect of the proteins (S1 + NC) combination was seen in the significant increase in IgG and IgA levels in both mucosa and serum. A recent study confirmed that NC + S1 intramuscular administration to mice showed an immunogenicity effect [28]. The findings in the present study confirm the effect of NC and S1 combination through an intranasal spray, which also provides evidence that formulating vaccines through nasal administration can effectively deliver effects to recipients.

In this study, we illustrated that nasal administration of low amounts of recombinant SARS-CoV-2 proteins combined with endogenous-like lipids as adjuvants induced primary and secondary immune responses in systemic IgG, mIL-5, and IFN-gamma secreting T lymphocytes, as well as humoral IgA in nasal and intestinal mucosal compartments in the majority of immunized mice. In this first attempt with the selected antigens, the number of animals responding with neutralizing antibodies in serum was not maximal and required the use of the cationic adjuvant N3. A significant increase in serum antibodies to the S1 spike + NC protein was observed in groups 5 and 6, which was formulated with the cationic N3 adjuvant. Similarly, the mucosal of the nasal respiratory tract and fecal IgA was significantly more strongly reactive in animals given antigens with an N3 adjuvant. This could be attributed to the destabilization mechanisms of cationic lipids on the cell membrane, which increase the permeability of the antigens and induces an immune response [29]. On other hand, an efficient concentration of the anionic L3 adjuvant is required to escape the cells membrane [30]. In this study, antigens with a relatively high concentration (2%) of the anionic L3-adjuvant emulsion administrated to mice showed a significant antibody response post the second immunization dose. The remaining amount of collected serum was used in vitro SARS-CoV-2 virus neutralization analysis. The data showed that the virus-neutralizing serum was only seen in groups 5 and 6 that received the cationic N3 adjuvant and the anionic L3-adjuvant group 4. However, significant neutralization was observed with the cationic adjuvants N3. These results illustrated that the functional antibody quality was superior when the N3 adjuvant was used.

Cell-mediated immunity, as investigated by the ELISpot analysis against the outer viral proteins (S1 spike and its cell-receptor-binding motif (S1-RBM) was seen, as well as against the more conserved internal NC. The main cellular immune response consisted of CD4+ T helper cells secreting murine IL-5 or INF-gamma, since the stimulating antigens used in the ELISpot assay have been shown to primarily contain MHC class II stimulating antigens. Practically no MHC class I cell-activation was shown. Significantly more cytokine secretion was seen when the cationic N3 adjuvant was used as an adjuvant, against all three SARS-CoV-2 representing antigens. Only modest anti-S1-protein-specific cell-mediated immune reactivity was seen if the anionic L3 adjuvant was used at the highest concentration.

Among prophylactic medical interventions against infectious diseases, vaccinations are the most potent and efficient preventive health care measures. The coronaviruses SARS, MERS, and SARS-CoV-2 have all three resulted in vaccine development, and the common protective immunity was antibody responses against the spike S1, S2, and E-proteins. For obtaining long-lasting immunity, helper-T cells (CD4+) and cytotoxic T cells (CD8+) are also needed. With the presented immunization procedure and vaccine antigens, a strong immune response was induced, immune-enhancing and supporting properties of the cationic adjuvant provided the highest levels of immunity, as have previously been shown with other vaccine antigens, such as influenza A and B, papillomavirus and HIV. The chosen adjuvants are mainly adapted for recombinant proteins and nucleic acids vaccines (plasmid DNA).

Results presented in this report introduce a broader conception of the coronavirus SARS-CoV-2 vaccination regimen. If the future respiratory tract viral infectious diseases, as it seems right now, we believe that vaccinations against this pandemic virus will require vaccine candidates that not only induce S1 spike protein direct immunity, but in the long run the vaccinees will be more beneficial if they develop immunity to some of the internal viral proteins as well. Therefore, we have included the nucleoprotein, the helical capsid (NC) which is the most frequent SARS-CoV-2 structural viral protein. The target was to provide neutralizing antibodies to the S1 spike outer protein and a cytotoxic T cell immunity to the inner viral NC-protein. A major advantage with this design would be to limit the loss of immune recognition due to mutations, such as in the newly evolving variants omicron [31]. The chances that mutations would allow immune escape within these proteins would be less likely as reported regarding the S1 spike mutants in the human population (i.e., the Brazilian P1, The British B.1.1.7, the South African or the Indian B.1.617.2 variants). These next generation and future vaccines should be able to provide broader viral targets more similar to the immunity obtained when recovering from a severe COVID-19 infection.

Within the research field behind the clinically EMA and FDA-approved SARS-CoV-2 S1-specific vaccines, several other SARS or MERS vaccine options have been investigated [5,32,33,34,35,36,37]. The difference with many of these candidates is their attempt to provide immunity to a larger number of viral proteins than the S1 Spike alone. The challenge is, however, to formulate the vaccine using the effective viral protein candidates that enhance protective immune responses; antibody-dependent enhancement (ADE) is favorable over other immunization options that could cause undesirable side effects.

Although wild-type C57BL/6 mice model is reported as not being a viable in vivo model for the SARS-CoV-2 infection, many studies have shown that common laboratory mice are susceptible to SARS-CoV-2 infection as well [38,39]. Chen Y and coworkers have recently shown the age-associated SARS-CoV-2 pathogenesis, immune responses, occurrence of re-infection and vaccine breakthrough infection using a wild-type C57BL/6 mouse model [40]. Different SARS-CoV-2 variants of concern are being reported to infect the respiratory tract and induce inflammatory response in wild-type laboratory mice [41]. Different variant of concerns are also using the wild-type murine ACE2 receptor for infection to replicate in the lungs [42]. Receptor-binding domain proteins of SARS-CoV-2 variants elicited robust antibody responses cross-reacting with wild-type and mutant viruses in mice [43]. Recently, Smith and co-workers have successfully shown the live COVID-19 virus neutralization and antigen-specific T cell responses by a DNA vaccine candidate for COVID-19 in C57BL/6 mice [44]. As previously discussed, this study focused mostly on the use of wild-type mice model in the further study of the immunogenic property of the vaccine and adjuvant within them. Thus, it definitely answered some important pre-challenge information of a wild-type animal model tool for the upcoming mouse-adapted SARS-CoV-2 strains.

In conclusion, the use of a nasally administered cationic N3-adjuvant emulsion containing the enveloped S1 spike and NC vaccine antigens enhanced a significantly broader adaptive immune response in a mouse model than those seen in non-adjuvanted antigens, or in a low-concentration anionic L3 adjuvant.

## Figures and Tables

**Figure 1 vaccines-10-00504-f001:**
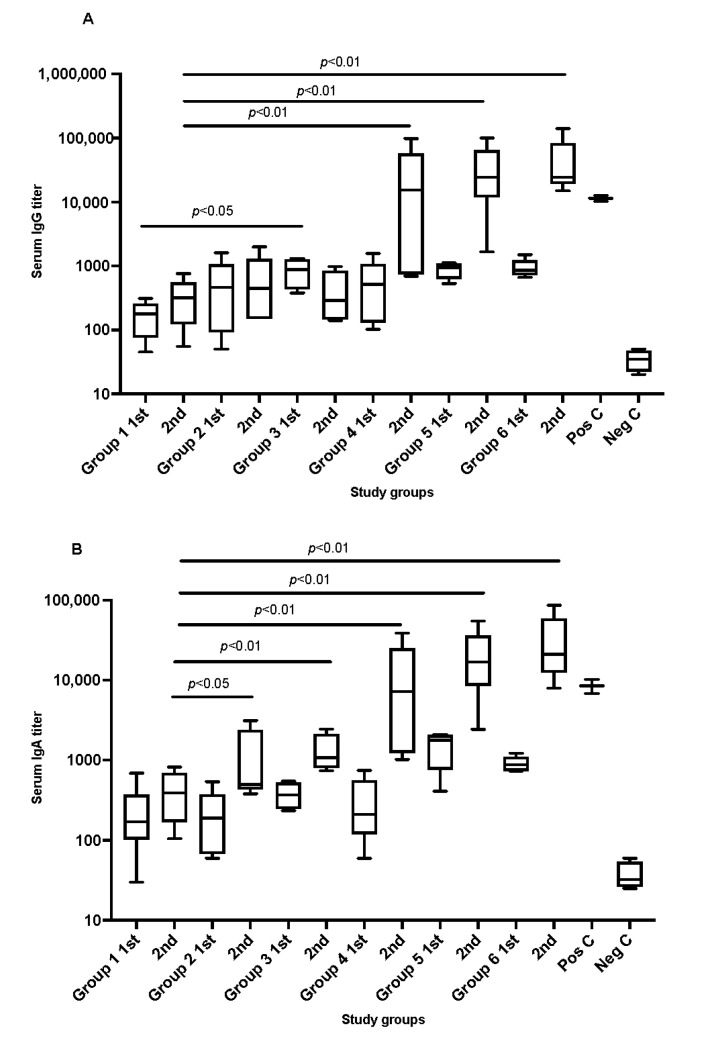
Graphs (**A**,**B**) shows the serum antibodies titration IgG and IgA, respectively, in immunized mice after administrating SARS-CoV-2 recombinant antigens in formulated vaccines. Serum was collected from each tested mouse group in week 3 and week 7 after administrating the recombinant proteins to the animals. Pos C is positive control and Neg C is the negative control group. Data are driven from 5 individuals per group, *p* value < 0.05.

**Figure 2 vaccines-10-00504-f002:**
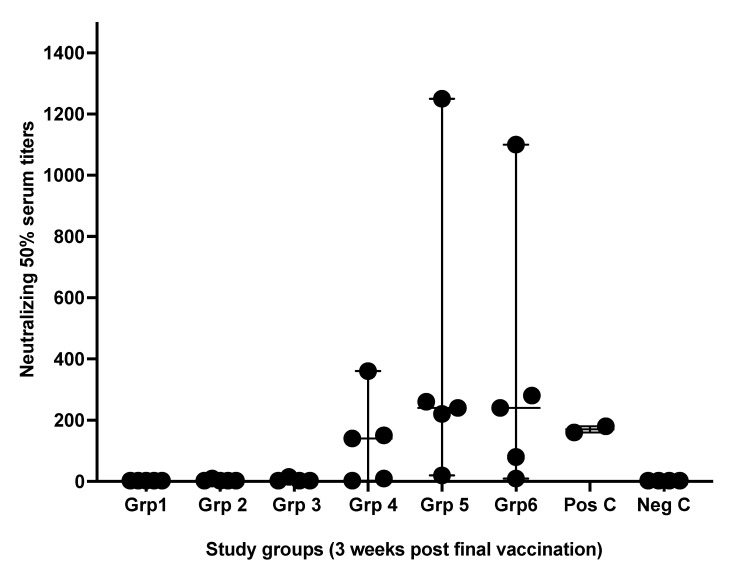
Microneutralizing serum antibodies titers. Serum antibody titers (NT) were obtained from mouse groups after three weeks of receiving the second immunizing dose of formulated vaccines. Serum antibodies levels were determined by endpoint neutralization of a 50% cell culture infectious and after 96 h of culture periods. Pos C is positive control while Neg C is the negative control group. Data are driven from 5 individuals per group, *p* value < 0.05.

**Figure 3 vaccines-10-00504-f003:**
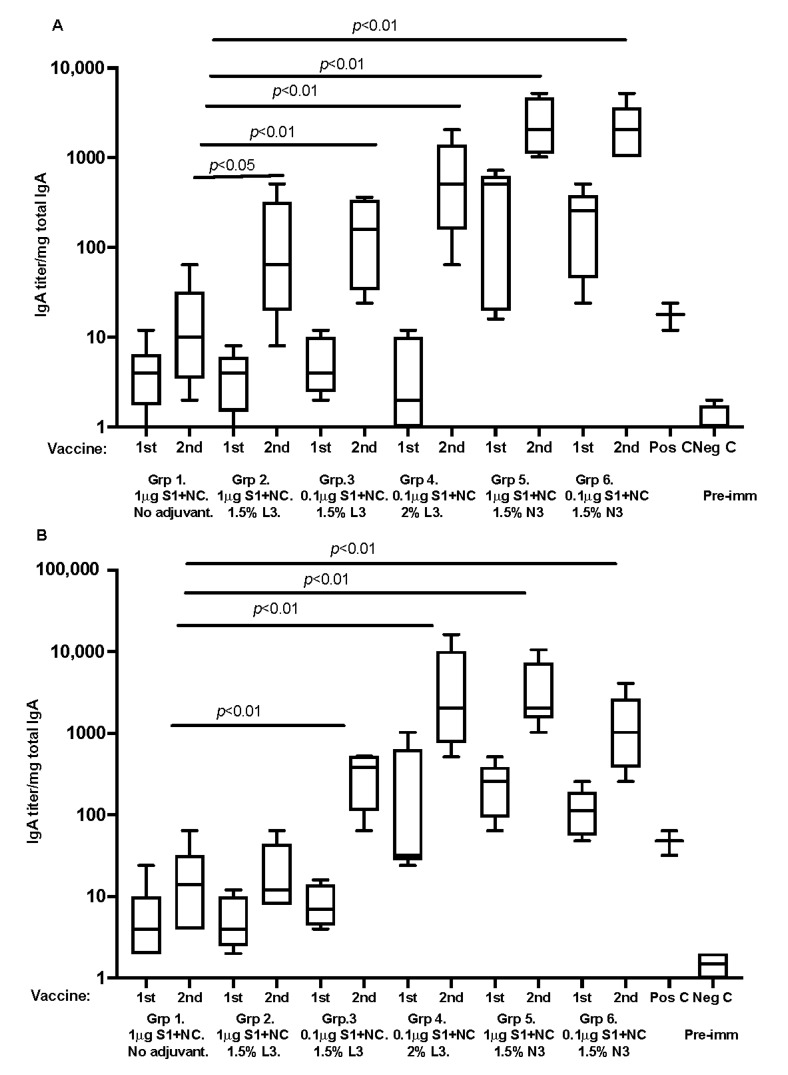
(**A**,**B**) graphs are representing respiratory tract lung-washed IgA antibody analyzed against S1 spike protein and (**B**) fecal pellet extracted IgA against S1 spike protein, respectively. Data are driven from 5 individuals per group, *p* value < 0.05.

**Figure 4 vaccines-10-00504-f004:**
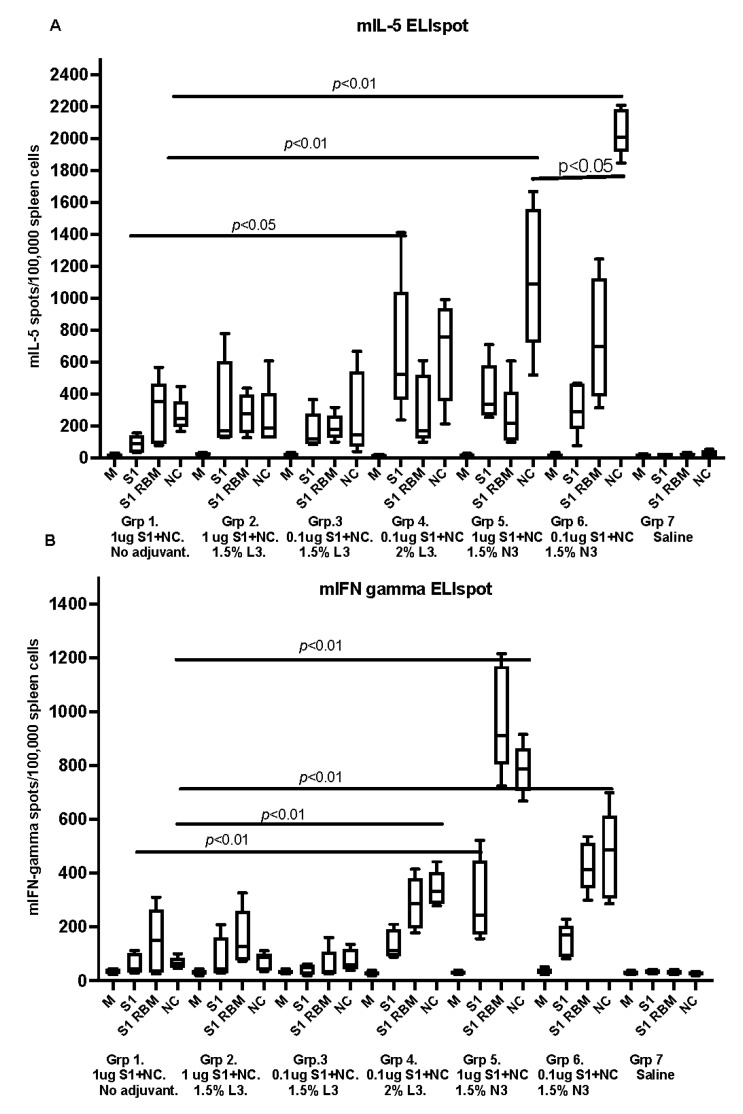
The graphs represent IL-5 and IFN-gamma secreted by spleen cells after vaccinating murine groups with the formulated vaccines. Graphs (**A**,**B**) show IL-5 and IFN-gamma secreted in blood collected on sacrificing day (post the second immunizing dose) and analyzed by ELIspot. Samples tested antigens used were cell culture medium (M) as negative control and 1 µg/mL of recombinant S1 spike protein, S1 receptor-binding motif antigen (S1 RBM) and 1 µg/mL nucleoprotein (NC) with cells from the respective study group. Data are driven from five individuals, *p* value < 0.05.

**Table 1 vaccines-10-00504-t001:** Illustrating research groups that introduced a certain vaccine formulation at a determined dose and adjuvant concentration. Abbreviations: L3 = Cationic lipid, N3 = Anionic lipid, NC = nucleocapsid, Pos C = positive control, Neg C = negative control.

Group	No.	Immunogens	Dose (µg)	Adjuvants
1	5	S1-Spike and NC	1	No
2	5	S1-Spike and NC	1	L3 1.5%
3	5	S1-Spike and NC	0.1	L3 1.5%
4	5	S1-Spike and NC	0.1	L3 2%
5	5	S1-Spike and NC	1	N3 1.5%
6	5	S1-Spike and NC	0.1	N3 1.5%
Pos C	3	Infected	0	No
Neg C	3	Saline	0	No

## Data Availability

Data can be made available upon individual request within a reasonable time.

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
