# Peer review of "Intranasal Coronavirus SARS-CoV-2 Immunization with Lipid Adjuvants Provides Systemic and Mucosal Immune Response against SARS-CoV-2 S1 Spike and Nucleocapsid Protein"

_vaccines, 2022, doi:10.3390/vaccines10040504_

Round 1

Reviewer 1 Report

This is an promising study that might lead to clinical studies.

Overall study is logical and provides an potentially  easier means of delivering vaccine if the human studies becomes available

It is reads nicely and I do not have any  major questions

Reviewer 2 Report

the paper appears well elaborated, however being a preliminary work it appears a bit long in content, some results could be present as supplementary material

Reviewer 3 Report

This study aims to explore the effect of the recombinant proteins SARS-CoV-2 (S1 spike and the nucleocapsid protein) to target different viral proteins. Moreover, a further goal is to determine the potential vaccine use and adjuvant-mediated enhancement of its efficacy in the wild-type mice model. Based on their results, the study demonstrated that nasal administration of low amounts of antigens/adjuvants induced a primary and secondary immune response.

The study is well written and the results well argued.

I have only a few suggestions in order to improve the discussion section.

A crucial point in the development of a new vaccine is represented by its safety. The authors missed discussing this important point: in this way, it could be useful this missed reference (DOI: 10.3390/jcm10245876) that summarized the fatal cases worldwide after COVID-19 vaccination. A new vaccine should be developed in order to avoid severe adverse effects. Please, insert a short paragraph about this theme.

Finally, the authors should insert the study's limitations.

Minor points:

- Line 115 there is a red word: please, check it.

- All names should be written univocally. please, check all text (i.e o sars cov2, SARS-CoV-2,....)